# Comparison of Mitotic Count and Ki-67 Index in Grading Gastroenteropancreatic Neuroendocrine Tumors and Their Association with Metastases

**DOI:** 10.3390/biomedicines13102445

**Published:** 2025-10-08

**Authors:** Mohammad Sheikh-Ahmad, Abed Agbarya, Sharon Talisman, Anan Shalata, Hadas Rabani, Jacob Bejar, Hila Kreizman Shefer, Reem Samara, Forat Swaid, Monica Laniado, Gideon Sroka, Nama Mubariki, Tova Rainis, Ilana Rosenblatt, Balsam Dakwar, Ekaterina Yovanovich, Leonard Saiegh

**Affiliations:** 1Institute of Endocrinology, Bnai Zion Medical Center, 47 Eliyahu Golomb Avenue, Haifa 3339419, Israel; anan.shalata@b-zion.org.il (A.S.); hadas.rabani@b-zion.org.il (H.R.); rosenblatt.ilana@gmail.com (I.R.); balsam.k.k@gmail.com (B.D.); rogachevsky26@gmail.com (E.Y.); leonard.saiegh@b-zion.org.il (L.S.); 2The Ruth and Bruce Rappaport Faculty of Medicine, Technion-Israel Institute of Technology, 1 Efron Street, Haifa 3339101, Israel; abed.agbarya@b-zion.org.il (A.A.); sharontalisman@gmail.com (S.T.); monica.laniado@b-zion.org.il (M.L.); gideon.sroka@b-zion.org.il (G.S.); tova.rainis@b-zion.org.il (T.R.); 3Institute of Oncology, Bnai Zion Medical Center, 47 Eliyahu Golomb Avenue, Haifa 3339419, Israel; 4Institute of Pathology, Bnai Zion Medical Center, 47 Eliyahu Golomb Avenue, Haifa 3339419, Israel; jacob.bejar@b-zion.org.il (J.B.); hila.shefer@b-zion.org.il (H.K.S.); reem.samara@b-zion.org.il (R.S.); 5Department of Surgery, Tzafon Medical Center, Poriya 1528001, Israel; fswaid@tzmc.gov.il; 6Department of Surgery, Bnai Zion Medical Center, 47 Eliyahu Golomb Avenue, Haifa 3339419, Israel; 7Division of Gastroenterology, Bnai Zion Medical Center, 47 Eliyahu Golomb Avenue, Haifa 3339419, Israel; naama.mabariki@b-zion.org.il

**Keywords:** neuroendocrine tumors, mitoses, Ki-67, metastases, WHO grading, prognostic markers

## Abstract

**Background**: Gastroenteropancreatic neuroendocrine tumors (GEP-NETs) are graded per the World Health Organization (WHO) using mitotic count and the Ki-67 index. There is an ongoing debate regarding the concordance between these parameters and their ability to predict metastatic disease. **Objective**: The objective is to assess concordance between the mitotic count and the Ki-67 index in grading GEP-NETs and to determine which parameter more accurately relates to metastatic disease and local tumor behavior. **Methods**: We conducted a single-center retrospective cohort study of adults with GEP-NETs managed between January 2006 and February 2024. Tumors were staged according to the TNM system. Grading followed WHO criteria using mitotic count and the Ki-67 index; when discordant, the higher grade was assigned. **Results**: Concordance between mitotic count- and Ki-67-based grading was 76.5% (78/102) with Cohen’s κ = 0.36, indicating fair-to-moderate agreement. More tumors were classified as G1 by mitotic count (86.3%) than by the Ki-67 index (68.6%). Neither mitotic count nor the Ki-67 index (numerical values or grades) showed a significant association with metastatic disease (all *p* > 0.05). Mitotic count (as a numerical continuous values) correlated with tumor invasion (T1 vs. T3, *p* = 0.035; T1 vs. T4, *p* = 0.036), whereas the Ki-67 index did not (*p* = 0.11). Tumor size was the strongest predictor of metastases (lymph-node *p* = 0.028; distant *p* < 0.001; any *p* < 0.001). **Conclusions**: Mitotic count and the Ki-67 index show only 76.5% concordance. Neither marker predicted metastatic disease in this cohort, while tumor size was the most robust predictor. These findings support giving greater weight to tumor size within prognostic algorithms while recognizing the limitations of proliferation-based grading for predicting metastasis.

## 1. Introduction

Gastroenteropancreatic neuroendocrine tumors (GEP-NETs) represent a heterogeneous group of malignancies arising from the diffuse neuroendocrine cell system throughout the gastrointestinal tract and pancreas. The incidence has increased 6.4-fold over nearly four decades, from 1.09 per 100,000 in 1973 to 6.98 per 100,000 in 2012 in the United States [1]. While the exact causes of this diagnostic increase remain uncertain, advances in imaging technology and enhanced awareness of neuroendocrine histological patterns probably contribute to this trend [2]. The World Health Organization (WHO) classification system serves as the gold standard for grading GEP-NETs [3]. This system divides well-differentiated neuroendocrine tumors into three grades (G1-G3) based on proliferative activity defined by mitotic count and the Ki-67 index. When discordance exists between these two parameters, the higher grade should be assigned [4].

The implementation of this dual-parameter approach has revealed significant challenges in clinical practice, particularly concerning the frequency of discordance between the Ki-67 index and mitotic count and their relative prognostic significance. These observations have led investigators to question whether both parameters contribute equally to prognosis or whether one might be superior for risk stratification [5,6].

McCall and colleagues, in a study involving 297 patients with pancreatic neuroendocrine tumors, found that grading using both the Ki-67 index and mitotic count increases prognostic accuracy compared to either parameter alone [7]. Their analysis revealed that discordance between the Ki-67 index and mitotic count was noted in 107 patients (36%). Crucially, they demonstrated that assigning the final grade based on the higher of the two parameters provided improved prognostic stratification and was a better predictor of survival than using either parameter independently [7]. Similarly, Philips and colleagues, in their multi-center study of 350 patients with pancreatic NENs, confirmed that discordance between the Ki-67 index and mitotic count was observed in 58 patients. Their multivariate analysis demonstrated that the combined final grade, based on the higher of the two parameters, was associated with superior hazard ratios for both overall survival and disease-free survival. Notably, the Ki-67 index proved to be a better predictor of overall survival, while mitotic count was more predictive of disease-free survival [8].

Khan and colleagues found discrepancies between the Ki-67 index and mitotic count in 44% and 38% of cases, during grading metastatic pancreatic and midgut neuroendocrine neoplasms, respectively [9]. In multivariate analysis, tumor grading based on the Ki-67 index, but not mitotic count, had a significant prognostic factor in determining overall survival. Moreover, the prognostic value of the Ki-67 index was improved using alternative Ki-67 index thresholds suggested by Scarpa et al. (G1: Ki-67 ≤5%, G2: Ki-67 >5% and ≤20%, G3: Ki-67 >20%) which yielded higher hazard ratios compared to the European Neuroendocrine Tumor Society (ENETS) Ki-67 index thresholds [4,10]. They concluded that the Ki-67 index alone should be used for grading pancreatic and midgut NENs and suggested that the current threshold for distinguishing G1 from G2 tumors should be revised from 2% to 5% [9].

The discordance between the Ki-67 index and mitotic count in predicting malignant behavior was clearly demonstrated by Lowe and colleagues, who studied 24 patients with pancreatic neuroendocrine tumors over a 5-year observation period [11]. Their findings revealed that while Ki-67 index ≥10% was 100% predictive of regional lymph node metastases, mitotic count ≥2/10 HPF was associated with lymph node spread in only 60% of cases. They also demonstrate that all tumors with Ki-67 index ≥10% had lymph node metastases compared to only 25% of tumors with Ki-67 index <10%. Additionally, during the observation period, mortality occurred in 80% of patients with a Ki-67 index ≥10%, whereas no deaths were observed among those with a Ki-67 index <10%. This finding suggests that higher Ki-67 thresholds may allow for more clinically meaningful risk stratification than conventional cut-offs and may provide superior prognostic accuracy compared with mitotic count alone [11].

These conflicting findings highlight a critical gap in our understanding of the utility of the Ki-67 index and mitotic count. Although studies have reported varying degrees of concordance between these two markers, fundamental questions remain regarding their true predictive value for metastatic disease. To address these uncertainties, we conducted a comprehensive analysis to assess parameter concordance and to determine which marker can reliably predict clinically meaningful outcomes in GEP-NETs.

## 2. Materials and Methods

Study Design and Patients

This single-center retrospective cohort study included patients diagnosed with GEP-NETs between January 2006 and February 2024. Patients were identified through a comprehensive review of medical records. Eligibility criteria required age ≥18 years at diagnosis, receipt of follow-up care at our institution, and a confirmed pathological diagnosis based on morphological features and immunohistochemistry of tissue obtained via biopsy, endoscopic resection, or surgical resection. Patients were excluded if clinical information was incomplete or pathological specimens were inadequate for grading assessment. All tumors were well-differentiated NETs; no poorly differentiated neuroendocrine carcinomas were included.

Data Collection and Definitions

Data collection involved a systematic review of electronic medical records, pathology reports, and imaging studies using standardized forms. Variables included patient demographics (age, gender), tumor characteristics (size, location, invasion), mitotic count and Ki-67 index numerical values and grade, treatment modalities (no resection, endoscopic resection, or surgical resection), and metastatic status. Metastatic status was categorized as: [1] no metastases, [2] regional lymph node metastases, [3] distant metastases or [4] any metastases, meaning the presence of either lymph node or distant metastasis. Tumor invasion was classified according to TNM staging (T1-T4). Pancreatic neuroendocrine tumors are staged by size and local extension; tumors confined to the pancreas and smaller than 2 cm are T1, those measuring 2–4 cm are T2, and tumors larger than 4 cm or invading the duodenum or bile duct are T3. Extension into adjacent organs such as the stomach, spleen, colon, or adrenal gland, or invasion of major vessels, defines T4. On the other hand, in the gastrointestinal tract, staging is based mainly on depth of invasion; tumors limited to the mucosa or submucosa up to 2 cm are T1. Invasion of the muscularis propria or tumors larger than 2 cm are T2. Extension through the muscularis into the subserosa or surrounding fat corresponds to T3, while perforation of the serosa or invasion of adjacent organs is classified as T4 [12]. Tumor size was determined by the maximal tumor diameter measured from pathological specimens or, when unavailable, from imaging studies. Imaging modalities included contrast-enhanced computed tomography (CT) in most cases and magnetic resonance imaging (MRI) in some. Radiology reports and images were reviewed to extract tumor measurements (when pathology was unavailable) and to ascertain metastases. Grading was performed according to WHO guidelines using both mitotic count and the Ki-67 index. WHO grade: G1 (<2 mitoses per 10 high-power fields (HPF) and Ki-67 index <3%), G2 (2–20 mitoses per 10 HPF and Ki-67 index 3–20%), and G3 (>20 mitoses per 10 HPF and Ki-67 index >20%). Ki-67 grade: G1 (Ki-67 index <3%), G2 (Ki-67 index 3–20%), and G3 Ki-67 index >20%). Mitotic grade: G1 (<2 mitoses per 10 HPF), G2 (2–20 mitoses per 10 HPF), and G3 (>20 mitoses per 10 HPF).

Pathology and Grading Assessments

Mitotic count was assessed on hematoxylin and eosin (H&E) stained tumor sections by counting mitotic figures across 50 HPF (400×, field area 0.237 mm^2^) in the most active area, then scaled to an average per 10 HPFs. Immunostains were performed using an automated stainer (Benchmark Ultra; Ventana Systems, Phoenix, AZ, USA). A Ki-67 index assessment was performed using immunohistochemistry with a mouse anti-human Ki-67 antibody (clone MIB-1, Dako, Glostrup, Denmark) with 32 min primary antibody incubation, evaluating 500–2000 cells in hot-spot areas. Both parameters were assigned grades according to WHO criteria. When discordance was present, the final grade was determined using the higher of the two individual grades in accordance with the WHO recommendations.

Statistical Analysis

Descriptive statistics (mean ± standard deviation, median, interquartile range, and percentages) were calculated for all study variables. Agreement between Ki-67 grading and mitotic grading was assessed using Cohen’s kappa with 95% confidence intervals. The normality of continuous variables was evaluated with the Kolmogorov–Smirnov test. Accordingly, between-group differences were examined using the Kruskal–Wallis test for the following comparisons: mitotic count level vs. T-stage; mitotic count level vs. metastases; tumor size vs. metastases; and Ki-67 level vs. metastases. For categorical/ordinal variables, Somers’ D was used. All tests were two-sided, and *p* < 0.05 was considered statistically significant. Statistical analyses were performed using SPSS version 29 (IBM Corp., Armonk, NY, USA).

## 3. Results

We screened 194 adults with GEP-NETs and, after excluding patients who did not meet prespecified inclusion criteria, analyzed a final cohort of 102 patients. The cohort’s mean age was 63.7 ± 14.6 years (range: 22–89) and nearly equal gender distribution (49 males, 53 females) (Table 1). Follow-up data were available for 96 patients, with a mean follow-up duration of 50.3 months ± 36.2 (range: 5–233.6). These data were used to determine whether metastases developed during the observation period.

In addition to resection and/or somatostatin analog therapy, two patients in this cohort received peptide receptor radionuclide therapy (PRRT), and one patient received everolimus. No patients received cytotoxic chemotherapy.

A significant discrepancy in grading was observed when classification was based on the Ki-67 index compared with the mitotic count (Table 2). Using the mitotic count, 88 patients (86.3%) were classified as G1, 13 patients (12.7%) as G2, and 1 patient (1.0%) as G3. In contrast, grading according to the Ki-67 index classified 70 patients (68.6%) as G1, 30 patients (29.4%) as G2, and 2 patients (2.0%) as G3. Among 102 cases, concordance analysis between the Ki-67 index grade and mitotic grade demonstrated agreement in 78 cases (76.5%) and disagreement in 24 cases (23.5%). The Cohen’s kappa coefficient was 0.36 (95% CI: 0.20–0.52), consistent with fair to moderate agreement (Table 2).

No correlation was observed between the Ki-67 index grade or Ki-67 numerical values and tumor T stage (r = 0.19, *p* = 0.11). In contrast, mitotic count numerical variable, but not mitotic grade, demonstrated a significant positive correlation with tumor T stage, particularly in comparisons of T1 vs. T3 (*p* = 0.035) and T1 vs. T4 (*p* = 0.036). (Table 3).

No correlation was found between Ki-67 numerical value or Ki-67 grade and the presence of metastases (*p* = 0.13 and *p* = 0.17, respectively). Similarly, no correlation was found between mitotic numerical value or mitotic grade and the presence of metastases (*p* = 0.095 and *p* = 0.06, respectively).

Tumor size emerged as the most robust predictor of metastatic disease in our GEP-NET cohort, showing highly significant correlations with lymph node metastases (*p* = 0.028), distant metastases (*p* < 0.001), and all metastases (*p* < 0.001) (Table 4). The mean tumor sizes increased progressively from 1.15 cm in non-metastatic cases to 4.81 cm in cases with distant metastases. On univariate logistic regression, each 1 cm increase in tumor size was associated with 2.64-fold higher odds of metastasis (95% CI 1.61–4.33, *p* < 0.001). In multivariable models, tumor size remained a strong predictor (OR 2.64, 95% CI 1.61–4.33, *p* < 0.001; and OR 2.62, 95% CI 1.59–4.32, *p* < 0.001), whereas mitotic count (OR 1.12, 95% CI 0.85–1.47, *p* = 0.44) and Ki-67 (OR 1.06, 95% CI 0.94–1.20, *p* = 0.34) were not statistically significant.

## 4. Discussion

Our study demonstrates a concordance rate of only 76.5% between mitotic count and the Ki-67 index based grading in GEP-NETs. This finding aligns with previous reports showing variable concordance rates ranging from 56–83% across different GEP-NETs [7,8,9].

The differences in grading distribution between mitotic count and the Ki-67 index deserve careful consideration. In our study, Ki-67 identified more G2 tumors (29.4%) compared to mitotic count (12.7%), with 20 of 24 discordant cases (83.3%) being upgraded from G1 by mitotic count to G2 by Ki-67.

The most striking finding of our study is the absence of correlation between either proliferation marker or metastatic disease. Neither mitotic count nor the Ki-67 index showed significant association with lymph node or distant metastases. This finding fundamentally challenges the current WHO grading paradigm, which positions proliferative activity as a key prognostic indicator [3,4].

Interestingly, while both markers failed to predict metastatic behavior, they showed disparity in assessing local tumor characteristics. Mitotic count numerical values demonstrated significant correlation with tumor T stage, well discriminating T1 from T3 and T4 tumors. In contrast, Ki-67 showed no correlation with T stage. This finding has potential clinical relevance for surgical planning and assessment of local recurrence risk.

In our cohort, tumor size emerged as the most powerful predictor of metastatic disease. The highly significant associations with lymph node metastases, distant metastases, and any metastases demonstrate a clear size-dependent metastatic risk. These findings are consistent with several recent studies emphasizing the prognostic importance of tumor size in GEP-NETs [13]. Bettini et al. demonstrated that tumor size > 2 cm was the strongest predictor of malignancy in nonfunctioning pancreatic NENs, independent of proliferation markers [14].

The pattern of discordance between the Ki-67 index and mitotic count reflects the fundamental biological differences between these markers: Ki-67 detects cells throughout most phases of the cell cycle (G1, S, G2, and M phases), while mitotic count captures only cells in active mitosis, which is the shortest phase of the cell cycle [15]. Additionally, the methodological emphasis on “hot spot” analysis for Ki-67 versus broader field assessment for mitotic count probably contributes to this difference [16].

The lack of standardization in identifying “hot spots” variations in antibody clones and dilutions, and differences in counting methodology all contribute to this discordance [17,18]. Moreover, interobserver variability in Ki-67 assessment has been well documented, with studies showing kappa values ranging from 0.32 to 0.68 even among experienced pathologists [17,18]. Recent advances in digital pathology and automated Ki-67 scoring show promise for improving reproducibility, though validation across different tumor types and institutions remains necessary [19,20,21].

Our findings have important implications for clinical practice and future research directions. First, the lack of correlation between proliferation markers and metastatic disease suggests that current WHO grading may be inadequate for predicting the most clinically relevant outcome-metastatic spread. This limitation reflects broader challenges with proliferation-based classification systems, which fail to capture the full biological complexity of neuroendocrine neoplasms. Recent investigations have identified potential molecular markers, including DAXX/ATRX alterations in well-differentiated tumors and RB1/TP53 mutations in poorly differentiated carcinomas, which may serve as adjunctive diagnostic tools [22,23]. The integration of morphological assessment, clinical presentation, imaging characteristics, and molecular profiling represents a promising approach toward more precise neuroendocrine tumor classification that could better predict metastatic potential and guide therapeutic decisions [13]. Second, the superior performance of tumor size in predicting metastases argues for its formal integration into prognostic algorithms, possibly as a primary stratification factor with proliferation markers serving a secondary role.

Study limitations include the retrospective single-center design and heterogeneous primary tumor sites. In addition, manual Ki-67 assessment carries an inherent limitation due to its variability. The relatively small number of G3 tumors prevented meaningful analysis of this important subgroup. Additionally, the absence of survival data limits our ability to assess the ultimate prognostic significance of these findings. Furthermore, tumor size measurements were obtained from pathology in resected cases and from imaging in unresected cases, which may introduce some measurement variability. Future prospective studies with standardized protocols, larger sample sizes, and long-term follow-up are needed to validate these observations and develop improved prognostic models.

## 5. Conclusions

This study found 76.5% concordance between mitotic count- and Ki-67-based grading of GEP-NETs. In this cohort, neither proliferation marker—considered as numerical continuous values or categorical grade—was significantly associated with the presence of metastases, whereas mitotic count (as numerical continuous values) correlated with local tumor invasion (T stage). Tumor size showed the strongest and most consistent association with metastatic disease. These findings support giving greater weight to tumor size within risk-stratification frameworks, alongside proliferation indices, while recognizing the limitations of proliferation-based grading for predicting metastasis. Validation in larger cohorts with standardized assessments and survival endpoints is warranted; in the interim, clinicians may consider tumor size as a key component of risk assessment and treatment planning.

## Figures and Tables

**Table 1 biomedicines-13-02445-t001:** Patient Demographics and tumor Characteristics.

Characteristic	Value	Percentage
Total patients	102	100%
Age (years), mean ± SD	63.7 ± 14.6	Range: 22–89
Gender:		
Male	49	48.0%
Female	53	52.0%
Primary tumor sites:		
Stomach	45	44.1%
Pancreas	22	21.6%
Appendix	10	9.8%
Rectum	8	7.8%
Small intestine	5	4.9%
Duodenum	6	5.9%
Unknown *	6	5.9%
Tumor size (cm), median [IQR]	1.05 [0.6–2.28]	Range: 0.1–10
WHO Grade:		
G 1	67	65.7%
G 2	33	32.4%
G 3	2	1.96%
Ki-67 grade:		
G 1	70	68.6%
G 2	30	29.4%
G 3	2	2.0%
Mitotic grade:		
G 1	88	86.3%
G 2	13	12.7%
G 3	1	1.0%
T stage (n = 71):		
T1	47	66.2%
T2	11	15.5%
T3	9	12.7%
T4	4	5.6%
Metastatic status (n = 96):		
No metastases	71	74.0%
Lymph node metastases	6	6.3%
Distant metastases	19	19.7%
Treatment modality:		
No resection	26	25.5%
Endoscopic resection	38	37.2%
Surgical resection	38	37.2%
Somatostatin analog treatment (n = 72):		
Yes	26	36.1%
No	46	63.9%

* Classified as GEP-NETs based on histopathological features. n = number of cases with available data.

**Table 2 biomedicines-13-02445-t002:** Concordance Analysis of Mitotic Count and Ki-67–Based Grading.

Grading	Ki-67 G1	Ki-67 G2	Ki-67 G3	Total
Mitotic G1	67 (76.1%) *	20 (22.7%)	1 (1.1%)	88
Mitotic G2	3 (23.1%)	10 (76.9%) *	0 (0%)	13
Mitotic G3	0 (0%)	0 (0%)	1 (100%) *	1
Total	70	30	2	102

* Cases with agreement between Mitotic Count and Ki-67–Based Grading.

**Table 3 biomedicines-13-02445-t003:** T Stage Distribution and Mitotic Count Correlation.

T Stage	N	Percentage	Median Mitotic Count Numerical Value (Range)	*p*-Value vs. T1
T1	47	66.2%	0 (0–8)	-
T2	11	15.5%	0 (0–5)	0.42
T3	9	12.7%	1 (0–11)	0.035
T4	4	5.6%	2 (0–40)	0.036

**Table 4 biomedicines-13-02445-t004:** Tumor Size and Metastatic Status.

Metastatic Status	N	Median Size (cm) [Range]	*p*-Value vs. no Metastasis
No metastasis	71	0.8 (0.1–4.5)	-
Lymph node	6	2.25 (1.5–3.0)	0.028
Distant	19	4.5 (0.55–10.0)	<0.001
Any metastasis	25	3.0 (0.55–10.0)	<0.001

## Data Availability

The data presented in this study is available upon request from the corresponding author.

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
