# Peer review of "Comparison of Mitotic Count and Ki-67 Index in Grading Gastroenteropancreatic Neuroendocrine Tumors and Their Association with Metastases"

_biomedicines, 2025, doi:10.3390/biomedicines13102445_

Round 1
Reviewer 1 Report
Comments and Suggestions for Authors
General Comments
This manuscript presents a retrospective cohort study evaluating the concordance between mitotic count and Ki-67 index in grading gastroenteropancreatic neuroendocrine neoplasms (GEP-NENs), as well as their association with metastatic disease. While the study addresses an important question, several methodological and interpretative limitations reduce the strength and generalizability of its conclusions.
Major Concerns
-
Prognostic Utility of Mitotic Count and Ki-67
The study concludes that neither mitotic count nor Ki-67 index significantly predicts metastatic disease. This finding challenges the rationale for using these markers as the foundation of grading systems. Despite 19 patients presenting with distant metastases, neither marker achieved statistical significance (Ki-67, p=0.13; mitotic count, p=0.095), suggesting limited prognostic value. -
Concordance and Reproducibility
Reported concordance between mitotic count and Ki-67 index is 76.5%, with a Cohen’s kappa of 0.36, reflecting only fair-to-moderate agreement. Of 102 cases, 24 showed discordant grading, and 83.3% were upgraded by Ki-67. This raises concerns about reproducibility and the potential for systematic risk inflation. -
Cohort Limitations and Generalizability
The single-institution, 18-year retrospective design introduces potential selection bias and limits external validity. Tumor site distribution is notably skewed (44.1% gastric origin), which may not reflect broader GEP-NEN epidemiology. -
Absence of Survival Endpoints
The lack of overall survival (OS) and disease-free survival (DFS) data significantly limits the clinical interpretability of grading discordance. Previous studies (e.g., McCall et al., Khan et al.) demonstrated survival implications of discordance, which this study cannot validate. -
Limited Representation of High-Grade Disease
Only 2 patients were classified as G3 by Ki-67 and 1 by mitotic count. This underrepresentation of aggressive tumors precludes meaningful assessment of grading performance in poorly differentiated disease. The authors’ conclusions regarding prediction of metastases may not extrapolate to higher-grade tumors.
Minor Concerns
-
Lack of detail on interobserver variability and digital quantification methods.
-
Potential limitations of manual Ki-67 scoring, given its known variability.
-
No description of validation or calibration across pathologists, despite citing variability in prior literature (e.g., Luchini et al., Bernhardt et al.).
-
Tumor size emerged as the strongest predictor of metastases, yet its integration into current grading systems is not explored.
-
Combining pancreatic, gastric, and intestinal NENs may mask site-specific grading behaviors.
-
Although DAXX/ATRX and TP53 mutations are mentioned, no molecular correlates were included, representing a missed opportunity for deeper risk stratification.
Author Response
Dear Reviewer,
Thank you very much for your thoughtful and constructive comments. We carefully addressed each of your points, and we believe the revisions have improved the manuscript.
Comment 1: Prognostic Utility of Mitotic Count and Ki-67
The study concludes that neither mitotic count nor Ki-67 index significantly predicts metastatic disease. This finding challenges the rationale for using these markers as the foundation of grading systems. Despite 19 patients presenting with distant metastases, neither marker achieved statistical significance (Ki-67, p=0.13; mitotic count, p=0.095), suggesting limited prognostic value.
Response: We agree with the reviewer that the absence of significant association with metastases suggests limited prognostic value of proliferation markers. This important limitation is acknowledged in the Discussion.
Comment 2: Concordance and Reproducibility
Reported concordance between mitotic count and Ki-67 index is 76.5%, with a Cohen’s kappa of 0.36, reflecting only fair-to-moderate agreement. Of 102 cases, 24 showed discordant grading, and 83.3% were upgraded by Ki-67. This raises concerns about reproducibility and the potential for systematic risk inflation.
Response: We agree with the reviewer that the level of concordance between mitotic count and Ki-67 (76.5%, κ=0.36) indicates only fair-to-moderate agreement, and that the predominance of Ki-67-driven upgrades in discordant cases is noteworthy. This pattern aligns with previously published observations and highlights the inherent variability of these markers.
Comment 3: Cohort Limitations and Generalizability
The single-institution, 18-year retrospective design introduces potential selection bias and limits external validity. Tumor site distribution is notably skewed (44.1% gastric origin), which may not reflect broader GEP-NEN epidemiology.
Response: We acknowledge the reviewer’s point regarding the limitations of a single-institution, long-term retrospective design. Such a framework inherently carries the risk of selection bias and reduces external generalizability. The predominance of gastric tumors (44.1%) reflects the referral and diagnostic patterns at our center and may not represent the broader epidemiology of GEP-NENs. Future multicenter studies with larger and more balanced cohorts will be needed to confirm the applicability of these findings.
Comment 4: Absence of Survival Endpoints
The lack of overall survival (OS) and disease-free survival (DFS) data significantly limits the clinical interpretability of grading discordance. Previous studies (e.g., McCall et al., Khan et al.) demonstrated survival implications of discordance, which this study cannot validate.
Response: We agree that the lack of overall survival and disease-free survival data limits the ability to fully interpret the prognostic significance of grading discordance. While previous studies demonstrated survival implications, our analysis was restricted to metastatic outcomes. Future studies with long-term survival follow-up will be necessary to validate these observations and place them in a broader clinical context.
Comment 5: Limited Representation of High-Grade Disease
Only 2 patients were classified as G3 by Ki-67 and 1 by mitotic count. This underrepresentation of aggressive tumors precludes meaningful assessment of grading performance in poorly differentiated disease. The authors’ conclusions regarding prediction of metastases may not extrapolate to higher-grade tumors.
Response: We agree with the reviewer that the very small number of G3 cases in our cohort does not allow for meaningful assessment of grading performance in poorly differentiated tumors. As such, conclusions regarding prediction of metastases cannot be generalized to higher-grade disease.
Comment 6: Lack of detail on interobserver variability and digital quantification methods.
Response: Each sample was examined by a single pathologist, and no digital quantification methods were available in our center.
Comment 7: Potential limitations of manual Ki-67 scoring, given its known variability.
Response: we have added in the section of limitations that: “In addition, manual Ki-67 assessment carries an inherent limitation due to its variability.” (Lines 276-277).
Comment 8: No description of validation or calibration across pathologists, despite citing variability in prior literature (e.g., Luchini et al., Bernhardt et al.).
Response: All samples were assessed by a single experienced pathologist, without parallel evaluation by a second observer. We acknowledge this as a methodological limitation and agree with the reviewer that lack of interobserver validation is an important caveat
Comment 9: Tumor size emerged as the strongest predictor of metastases, yet its integration into current grading systems is not explored.
Response: We appreciate this comment regarding tumor size. In our dataset, tumor size indeed emerged as the strongest predictor of metastatic spread, which is consistent with known behavior in NETs. We have performed univariate and multivariate analyses to evaluate whether tumor size independently predicts metastatic disease. Using our dataset, we conducted logistic regression modeling with metastasis (yes/no) as the outcome. The univariate analysis confirmed that larger tumor size is associated with higher odds of metastasis. Importantly, in a multivariate logistic regression including tumor size, Ki-67 index, and mitotic count simultaneously, tumor size remained the only significant independent predictor of metastatic disease. Neither Ki-67 nor mitotic count was significant in the multivariate model, suggesting that tumor size’s predictive value is not attributable to confounding by these proliferation markers. We have added these results to the manuscript, using standard statistical terminology (odds ratios and confidence intervals) to report the findings (Lines 215-220).
Comment 10: Combining pancreatic, gastric, and intestinal NENs may mask site-specific grading behaviors.
Response: We acknowledge the potential heterogeneity introduced by grouping different primary sites together. Because of the sample size constraints, detailed site-specific analysis was limited. We agree that future studies with larger sub-cohorts should examine site-specific patterns separately.
Comment 11: Although DAXX/ATRX and TP53 mutations are mentioned, no molecular correlates were included, representing a missed opportunity for deeper risk stratification.
Response: We agree that the inclusion of molecular data would provide valuable additional insights. Unfortunately, molecular correlates (e.g., specific gene mutations or expression profiles) were beyond the scope of our retrospective study and such data were not available for our cohort.
Reviewer 2 Report
Comments and Suggestions for Authors
Comments on "Comparison of Mitotic Count and Ki-67 Index in Grading Gastroenteropancreatic Neuroendocrine Tumors and Their Association with Metastases" by Sheikh-Ahmad et al.
In this manuscript, authors analyzed 102 adults with GEP-NENs, managed between January 2006 and February 2024, to assess concordance between mitotic count and Ki-67 index in grading GEP-NENs and to determine which parameter more accurately relates to metastatic disease and local tumor behavior. Although the topic is of interest, there are several inconsistencies that need to be addressed. Please see my comments appended below:
1. Details of mitotic count and Ki-67 index are word-to-word repeated in Introduction, Methods, and Results section. Methods section is most appropriate for this information.
2. Methods section should be renamed as Materials and Methods section. Authors mention a few materials such as antibodies, sources of these materials should be provided.
3. Methods section should be split into appropriate subsections.
4. Line 117: How is category "(4) any metastases" different from other 3 categories?
5. Lines 127-129: "Tumor size was determined by the maximal tumor diameter measured from pathological specimens or, when unavailable, from imaging studies." Authors should use only one modality for consistency or present concordance between the estimation of tumor size using the two modalities.
6. Authors extract data from imaging studies. Details of imaging studies should be mentioned.
7. How mitotic count was determined? Was any staining involved? Give details.
8. Lines 143-145: Aim of the study should not be part of Methods section.
9. Instead of using an umbrella of GEP-NENs, authors must adopt WHO classification 2022 and distinguish GEP-NETs from GEP-NECs. According to the Table 1 patients are of GEP-NETs. Thus, the terminology GEP-NENs is misleading.
10. Table 1: Did patients get any other therapy other than resection and/or SSA treatment, post resection?
11. Authors mention about the follow-up data, but it is not clear where is it used?
12. All 102 patients underwent resection (Table 1). 71 patients had no metastasis. In remaining patients were metastases present at diagnosis? Or developed during the follow-up period?
13. The title of the article is "Comparison of Mitotic Count and Ki-67 Index in Grading Gastroenteropancreatic Neuroendocrine Tumors and Their Association with Metastases". Authors show that "Tumor size emerged as the most robust predictor of metastatic disease." This is the most important finding of this article. The data should be presented in a table.
14. Authors must test whether tumor size is an independent predictor of metastatic disease through univariate and multivariate analyses.
Author Response
Dear Reviewer,
Thank you very much for your thoughtful and constructive comments. We carefully addressed each of your points, and we believe the revisions have improved the manuscript.
Comment 1: “Details of mitotic count and Ki-67 index are word-to-word repeated in Introduction, Methods, and Results section. Methods section is most appropriate for this information.”
Response: We agree with the reviewer and have removed redundant descriptions of the grading criteria from the Introduction and Results. The detailed definitions of mitotic count and Ki-67 index (and their grade cut-offs) are now presented only in the Methods section.
Comment 2: “Methods section should be renamed as Materials and Methods section. Authors mention a few materials such as antibodies, sources of these materials should be provided.”
Response: We have retitled the Methods section to “Materials and Methods” in accordance with journal style. Additionally, we have now specified the sources and details for materials used. In particular, the Ki-67 immunohistochemistry antibody is now described with its clone and manufacturer to ensure transparency.
Comment 3: “Methods section should be split into appropriate subsections.”
Response: We concur and have organized the Materials and Methods into clear subsections for readability. These subsections separate the study design, data collection, pathological assessments, and statistical analysis.
Comment 4: “Line 117: How is category ‘(4) any metastases’ different from other 3 categories?”
Response: The category “(4) any metastases” was intended as a composite category to indicate the presence of any metastatic disease (either regional nodal or distant), in contrast to category (1) which indicates no metastasis. We realize this needed clarification. We have now explicitly defined “any metastases” as a separate binary outcome used for analysis (presence of any metastasis vs none). This composite category groups (2) and (3) together to facilitate certain comparisons.
In the Materials and Methods description of metastatic status, we have revised the text to: “Metastatic status was categorized as: (1) no metastases; (2) regional lymph node metastases; (3) distant metastases. For some analyses, we also defined (4) ‘any metastases,’ meaning the presence of either lymph node or distant metastasis.” This clarifies the purpose of category 4. (Line 122).
Comment 5: “Lines 127-129: ‘Tumor size was determined by the maximal tumor diameter measured from pathological specimens or, when unavailable, from imaging studies.’ Authors should use only one modality for consistency or present concordance between the estimation of tumor size using the two modalities.”
Response: We acknowledge the concern about mixing measurement modalities. Ideally, a single modality would be used for consistency; however, in our retrospective cohort, 26 patients did not undergo resection, so pathological tumor measurements were unavailable for them. Thus, we used imaging-derived size for those cases to avoid excluding data. We have now clarified this in the Methods and addressed the potential inconsistency. Unfortunately, because each tumor’s size was measured by only one modality (whichever was available), we could not directly compare pathology vs imaging measurements for the same tumor. We have added a note about this limitation. (Lines 279-281).
Proposed Revision: In Materials and Methods, we now write: “Tumor size was recorded as the maximal diameter of the tumor from the pathology specimen for resected tumors. For patients who did not undergo resection, the maximal tumor diameter was obtained from imaging studies”. Additionally, in the Discussion (Study Limitations), we include: “Furthermore, tumor size measurements were obtained from pathology in resected cases and from imaging in unresected cases, which may introduce some measurement variability.”
Comment 6: “Authors extract data from imaging studies. Details of imaging studies should be mentioned.”
Response: We have expanded the Methods to specify the imaging modalities and how imaging data were used. In our cohort, most patients underwent cross-sectional imaging (contrast-enhanced CT scans being most common, with MRI in select cases) as part of their diagnostic workup. We have stated that radiology reports (and images when needed) were reviewed to determine tumor size (for unresected cases) and to identify metastatic lesions. (Lines 134-137).
Comment 7: “How was mitotic count determined? Was any staining involved? Give details.”
Response: The mitotic count was determined by manual counting on standard hematoxylin and eosin (H&E)-stained histopathology slides. No special immunohistochemical stain was used for mitoses. We have clarified in the Methods that a pathologist counted the number of mitotic figures in 50 high-power fields (in the most mitotically active tumor region) on H&E slides, and then calculated the average per 10 HPF for grading. (Lines 144-146).
Comment 8: “Lines 143-145: Aim of the study should not be part of Methods section.”
Response: We agree that the statement of the study’s aim does not belong in the Methods. We have removed the sentence stating our research aim from the Methods section.
Comment 9: “Instead of using an umbrella of GEP-NENs, authors must adopt WHO classification 2022 and distinguish GEP-NETs from GEP-NECs. According to Table 1 patients are of GEP-NETs. Thus, the terminology GEP-NENs is misleading.”
Response: We appreciate this important clarification. All patients in our series had well-differentiated gastroenteropancreatic neuroendocrine tumors (GEP-NETs) – none were poorly differentiated neuroendocrine carcinomas (NECs). We have revised the terminology throughout the manuscript to align with the 2022 WHO classification. Specifically, we now refer to our cases as GEP-NETs (neuroendocrine tumors) rather than GEP-NENs (neuroendocrine neoplasms), to reflect that these were well-differentiated tumors. We also explicitly state that no GEP-NECs were included. (Lines 113-114).
Comment 10: “Table 1: Did patients get any other therapy other than resection and/or SSA treatment, post resection?”
Response: we added to the article: In addition to resection and/or somatostatin analog therapy, two patients in this co-hort received peptide receptor radionuclide therapy (PRRT), and one patient received everolimus. No patients received cytotoxic chemotherapy. (Lines 186-188).
Comment 11: “Authors mention about the follow-up data, but it is not clear where is it used?”
Response: We have clarified how follow-up data were utilized. Follow-up information (available for 96 of 102 patients) was mainly used to determine longitudinal outcomes, particularly the development of metastases over time. In our analysis, a patient’s metastatic status was considered positive if metastases were present either at diagnosis or during the follow-up period. We did not, however, perform separate survival analyses or time-to-event analyses in this study. To avoid confusion, we have now explicitly stated the role of follow-up data in assessing metastasis. (Lines 178-179).
Comment 12: “All 102 patients underwent resection (Table 1). 71 patients had no metastasis. In remaining patients were metastases present at diagnosis? Or developed during the follow-up period?”
Response: Among the patients with metastatic disease, some had metastases already present at diagnosis, while others developed metastases during follow-up. We did not differentiate between these groups in the analyses, as this distinction would not alter the conclusions of the study.
Comment 13: “The title of the article is ‘...Association with Metastases.’ Authors show that ‘Tumor size emerged as the most robust predictor of metastatic disease.’ This is the most important finding of this article. The data should be presented in a table.”
Response: We agree that the relationship between tumor size and metastatic spread is a key finding and merits a dedicated table. We have created a new table (now Table 4 in the manuscript, line 209) that presents tumor size statistics stratified by metastatic status. This table clearly shows how tumor size differed between patients with and without metastases, highlighting the strong correlation. It includes the median tumor sizes for non-metastatic cases, cases with lymph node metastases, distant metastases, and any metastasis, along with the relevant p-values for comparisons.
Comment 14: “Authors must test whether tumor size is an independent predictor of metastatic disease through univariate and multivariate analyses.”
Response: We have performed the requested univariate and multivariate analyses to evaluate whether tumor size independently predicts metastatic disease. Using our dataset, we conducted logistic regression modeling with metastasis (yes/no) as the outcome. The univariate analysis confirmed that larger tumor size is associated with higher odds of metastasis. Importantly, in a multivariate logistic regression including tumor size, Ki-67 index, and mitotic count simultaneously, tumor size remained the only significant independent predictor of metastatic disease. Neither Ki-67 nor mitotic count was significant in the multivariate model, suggesting that tumor size’s predictive value is not attributable to confounding by these proliferation markers. We have added these results to the manuscript, using standard statistical terminology (odds ratios and confidence intervals) to report the findings. (Lines 215-220).
Round 2
Reviewer 1 Report
Comments and Suggestions for Authors
The authors have addressed all concerns satisfactorily.
Reviewer 2 Report
Comments and Suggestions for Authors
Authors have addressed all the comments.